# A Comprehensive Characterization of Small RNA Profiles by Massively Parallel Sequencing in Six Forensic Body Fluids/Tissue

**DOI:** 10.3390/genes13091530

**Published:** 2022-08-25

**Authors:** Zhiyong Liu, Qiangwei Wang, Nana Wang, Yu Zang, Riga Wu, Hongyu Sun

**Affiliations:** 1Faculty of Forensic Medicine, Zhongshan School of Medicine, Sun Yat-sen University, Guangzhou 510080, China; 2Guangdong Province Translational Forensic Medicine Engineering Technology Research Center, Sun Yat-sen University, Guangzhou 510080, China

**Keywords:** forensic stains, miRNA, piRNA, snoRNA, unique molecular identifiers, massively parallel sequencing

## Abstract

Body fluids/tissue identification (BFID) is an essential procedure in forensic practice, and RNA profiling has become one of the most important methods. Small non-coding RNAs, being expressed in high copy numbers and resistant to degradation, have great potential in BFID but have not been comprehensively characterized in common forensic stains. In this study, the miRNA, piRNA, snoRNA, and snRNA were sequenced in 30 forensic relevant samples (menstrual blood, saliva, semen, skin, venous blood, and vaginal secretion) using the BGI platform. Based on small RNA profiles, relative specific markers (RSM) and absolute specific markers (ASM) were defined, which can be used to identify a specific body fluid/tissue out of two or six, respectively. A total of 5204 small RNAs were discovered including 1394 miRNAs (including 236 novel miRNA), 3157 piRNAs, 636 snoRNAs, and 17 snRNAs. RSMs for 15 pairwise body fluid/tissue groups were discovered by differential RNA analysis. In addition, 90 ASMs that were specifically expressed in a certain type of body fluid/tissue were screened, among them, snoRNAs were reported first in forensic genetics. In brief, our study deepened the understanding of small RNA profiles in forensic stains and offered potential BFID markers that can be applied in different forensic scenarios.

## 1. Introduction

Body fluids (venous blood, menstrual blood, saliva, semen, and vaginal secretion) and skin tissue are frequently encountered biological materials at a crime scene. Therefore, the identification of the origin of different body fluids/tissue is indispensable for reconstructing the offense scene and providing scientific clues or evidence. Usually, forensic body fluid identification (BFID) is an essential forensic step besides DNA-based individual identification [1]. The traditional enzymatic/immunological and morphological tests are still necessary for their convenience and celerity, however, the sensitivity is low and sample consumption is high [2,3], restricting their applications in practice. Molecular-level analysis such as DNA methylation [4,5], microbiome [6], mRNA [7,8], miRNA [9], piRNA [10], etc., have also been explored by forensic scientists for forensic BFID. Among them, mRNA seems to be a promising BFID marker and has been researched widely. However, mRNA is susceptible to degradation by various ribonucleases or environmental factors, restricting its large-scale applications in BFID [11].

The miRNA are small (~22 nt) non-coding single-stranded RNAs that have tissue-specific expression and are also present in various extracellular body fluids such as blood, saliva, urine, and breast milk [12]. It exists in eukaryotic organisms broadly and plays an essential role at the post-transcriptional level in regulating gene expression through both translational inhibition and mRNA destabilization [13]. In forensics, miRNA has been proposed as novel and potential forensic BFID markers [14] because of its abundance and tissue-specific expression. Not only that, due to their molecular structure, miRNAs can resist degradation by environmental factors (e.g., heat, humidity or UV light) [15] or chemical factors [16], showing stronger stability than mRNA. These advantages make the application of miRNA in forensic practice possible, and some miRNA-based body fluids/tissue identification systems have been developed [17,18].

The wide use of massively parallel sequencing (MPS) has greatly advanced the discovery of small noncoding RNAs (sncRNAs). In addition to miRNA, the PIWI-interacting RNA (piRNA), small nucleolar RNA (snoRNA), and small nuclear RNA (snRNA) also belong to sncRNA (<300 nt in length) [19], and have attracted increasing interest from forensic geneticists. The piRNA is single-stranded ncRNA consisting of 21–35 nucleotides [20], performing an essential function by interacting with P-element-induced wimpy testis (PIWI) proteins during spermatogenesis [21]. The snoRNA [22], the most abundant group of intron-encoded ncRNAs, is predominantly found in the nucleolus, though may also exist in Cajal bodies or nucleoli where they assist the ribosomal RNA in their post-transcriptional modifications [23,24]. The snRNAs are non-polyadenylated, highly abundant sncRNAs that localize in the nucleus with important functions in intron splicing and other RNA processing [22,24]. According to the common sequence features and protein cofactors, it is often divided into two classes, sm-class snRNA (U1, U2, U4, U4atac, U5, U7, U11, and U12) and lsm-class snRNA (U6 and U6atac) [24].

Although plenty of studies have been conducted to investigate sncRNAs for BFID, most were focused on miRNA and a comprehensive profile of the sncRNAs, which targets different types of sncRNAs such as piRNA, snoRNA, and snRNA, are rare. Nevertheless, a comprehensive characterization of the RNA profile is the first step in selecting markers for BFID. At present, sncRNAs markers discovered through different BFID studies have often been inconsistent [25,26,27]. Hence, a systematic characterization and evaluation of these small RNAs in forensic biological materials will be highly valuable. In this study, the small RNA profiles of 30 samples (five body fluids and one skin tissue) were characterized. To accurately quantify the small RNA expression level in MPS, we adopted a purification strategy by electrophoretic separation for destination small RNA segments from total RNAs before the library construction. It was a direct and effective method for obtaining the initial input of small RNAs. Meanwhile, to decrease the influences of amplification and sequencing errors, unique molecular identifiers (UMI) were used. The small RNA profiles that showed diversity among the different groups and different sncRNA markers for body fluids/tissue identification were selected.

## 2. Materials and Methods

### 2.1. Body Fluids and Skin Tissue Sampling

First, five kinds of forensically related fresh body fluids and skin tissue were collected from healthy individuals. Informed consent was obtained from all of the volunteers. Five people were sampled for each type of sample. Semen (SE) and saliva (SA) were collected in 5 mL sterile tubes and put in a −80 °C refrigerator quickly after liquid nitrogen freezing. Menstrual blood (MB; the first three days) and vaginal secretion (VS) were collected with a sterile flocking swab (Huachenyang, Shenzhen, China). The skin (SK) was collected by rubbing the left forearm with a sterile water-wetted flocking swab. All swabs were put in the preservation solution (Huachenyang) temporarily, and then in a −80 °C refrigerator as soon as possible. Venous blood (VB) was collected from elbow veins in a 5 mL EDTA anticoagulant tube. To decrease the influence of heme from red blood cells on subsequent sequencing, human lymphocyte separation liquid (MD Pacific, Tianjin, China) was used to separate the leukocyte from venous blood, which was then stored in TRIzol reagent (Thermo Fisher Scientific, Waltham, MA, USA) and put in a −80 °C refrigerator. The information of all samples is shown in Appendix A. A total of 30 samples were sequenced including five people for each of the six types of samples, and an extra technical repetition VB006, which was identical to VB002. The repetition was internal quality control and not included in further VB group-related analysis. This study was approved by the Ethics Committee of Zhongshan School of Medicine, Sun Yat-sen University (No. [2022] 026).

### 2.2. RNA Extraction and Quantification

The total RNA was extracted from five body fluids and skin tissue using Trizol (Invitrogen, Carlsbad, CA, USA) according to the manufacturer’s instructions. RNA integrity number (RIN) and quantity were determined using Agilent RNA 6000 Pico Reagents with Agilent 2100 Bioanalyzer (Agilent Technologies, Santa Clara, CA, USA). The RIN value (1–10) is an index to determine the degradation degree of RNA, and the smaller the value, the more severe the degradation. Usually, the RIN of eight is recommended for MPS library construction. A NanoDrop (Thermo Fisher Scientific) was used to detect the inorganic ions or polycarbonate contamination.

### 2.3. Small RNA Library Preparation and Sequencing

The RNA library was prepared with 200 ng~1 μg of input RNAs. To extract small RNAs, purification was performed using electrophoretic separation on a 15% denaturing urea polyacrylamide gel electrophoresis (PAGE) gel. The 18–30 nt gel bands that corresponded to small RNAs were purified. Then, we ligated the small RNAs to adenylated 3′ adapters annealed to unique molecular identifiers (UMI) (Sangon Biotech, Shanghai, China), followed by the ligation of 5′adapters. After transcribing the adapter-ligated small RNAs into cDNA by SuperScript II Reverse Transcriptase (Invitrogen), we enriched the resulting fragments by PCR amplification with the PCR Primer Cocktail and PCR Mix (Sangon Biotech, Shanghai, China). Finally, the targeted 110~130 nt PCR products were selected by agarose gel electrophoresis using a QIAquick Gel Extraction Kit (QIAGEN, Valencia, CA, USA). We quantified the purified libraries using real-time quantitative PCR (qPCR; TaqMan Probe) and qualified them by examining the distribution of the fragment sizes with an Agilent 2100 bioanalyzer. Sequencing was performed with the BGISEQ-500 platform (BGI, Shenzhen, China) with the single-end (SE) mode.

### 2.4. Bioinformatics Analysis

Raw sequencing data, or raw reads, were processed with FastQC (Bioinformatics Group at the Babraham Institute, Cambridgeshire, UK) to eliminate low-quality reads such as reads with 5′ primer contaminants or poly-A, reads without 3′ primer, and reads shorter than 18 nt. The clean reads were mapped to the reference genome (GCF_000001405.38_GRCh38.p12) and small RNA databases such as miRbase (miRNA) [28], piRNABank (piRNA) [29], and snoRNABase (snoRNA) [30] with Bowtie2 [31] using default parameters. Particularly, cmsearch [32] was conducted for Rfam database mapping. The tRNA and rRNA were deleted. When putative miRNAs could not be found in the above database, the miRDeep2 software was used to predict novel miRNA according to previous studies [33]. The raw expression level of small RNA was calculated by counting the absolute numbers of molecules using UMI [34]. Furthermore, they were standardized as EXP values with RSEM (v1.2.12) (https://github.com/deweylab/RSEM, accessed on 23 April 2021), an accurate and user-friendly software tool for quantifying the transcript abundances from RNA-Seq data. The analysis of differential expression was performed using DEGseq2 [35] with default thresholds (*q*-value < 0.05 and the absolute value of Log_2_ Ratio >1 (two folds). The reference genes were screened by ERgene using default parameters [36].

### 2.5. Body Fluids/Tissue-Specific RNA Analysis

In our search, we defined two categories of body fluids/tissue-specific markers, the relative specific markers (RSM) and absolute specific markers (ASM). The RSM means that the markers are specific to some of the body fluids/tissue combinations. It could be used to pick out the interested body fluid/tissue within the assumed pairwise combinations and was not for the purpose of picking one fluid/tissue type out of six types of stains, while ASM is a marker that is specifically expressed in a certain type of body fluid/tissue, which is supposed to be able to help pick one fluid/tissue type out of common forensic stains (six types in this study). According to forensic practice, these two types of markers have different usage scenarios. For instance, to reconstruct the crime scene in a rape case that may involve a mixture of semen and vaginal secretion, RSM markers that could identify semen from a semen and vaginal secretion mixture rather than ASM markers that identify semen out of the six stains are enough to use. Based on the concepts, the RSMs were screened first. For the identification of RSMs, we set the criteria as the expression changes of 2-fold change (FC), in other words, log_2_FC more than 1, and a *q*-value (adjusted *p*-value) less than 0.05 in the pairwise groups. Then, for the identification of ASMs, the criteria of log_2_FC of more than 1 when the expression was compared with any of the other five groups and a *q*-value less than 0.05 were applied.

## 3. Results and Discussion

### 3.1. Small RNA Sequencing Results

A summary of RNA concentration and RIN of the total RNA are listed in Table 1. In six groups, the VS group had the highest mean RNA concentration (270.60 ng/μL), while SK showed the lowest mean RNA concentration (1.90 ng/μL). For RIN, the values of the VB group were within the range of 9.3–9.7 (Appendix A), meeting the recommended RIN value (>8) for library construction, while that of the menstrual blood, semen, saliva, vaginal secretion, and skin consistently exhibited lower scores (RIN: 1.1–2.6; Figure 1A–C).

After filtering, the total mean clean reads ranged from 21.11 M to 25.38 M (Table 1), with an average of 23.97 M per sample (Appendix A). The group with the lowest average Q20 reached 98.76% (Table 1). The read number distribution of different fragment sizes of the small RNA is an important feature of sequencing quality. As shown in Figure 2, the read counts of the clean segments were in the 1.00 × 10^6^, which fluctuated up and down in most samples. However, some samples such as SK004, MB005, and VB001 exhibited relatively unbalanced sequencing depth among the different clean reads.

Then, all of the clean reads were aligned to the reference databases and the mapping percentages were calculated for all of the samples (Appendix A). As shown in the table, the SE and VB group showed a high matching rate (~90%, besides VB003: 81.3%), followed by the MB and VS groups with a mean matching rate of 77.34 and 77.94%, respectively (Table 1). However, the mean mapping percentages of the SA (29.31%) and SK (33.90%) groups were lower and more variable within the group (Appendix A). The low human RNA or high microbiome RNA could be one of the reasons [37]. In our study, the RIN values of five groups of samples (excluding VB group, mean RIN: 9.52) were low, which indicated that the RNAs were “highly degraded”. Nevertheless, the data quantity (clean read count) and quality (Q20) obtained was high enough for subsequent analysis, which was also observed in a previous study [38]. In terms of the mean mapping rate, the SE group was the highest, followed by the VB group, while SA showed the poorest result (Table 1). Perhaps the variation is mainly dependent on the properties of the sample itself. As shown in Figure 1A–C, the RIN values did not seem to be relevant to the RNA concentrations, read counts, and mapping rates. Or in other words, the RIN index did not seem to affect the real sequencing quality of the small RNA that much. Some samples with lower RIN could still obtain enough data, especially the semen samples. One of the reasons is that RIN value may represent the overall quality of the RNA fragments, which includes both large and small RNAs and the small RNAs are the ones with higher anti-degradation (bio/env) properties [39,40,41]. In a way, the small RNAs did not seem to be susceptible to degradation, and they could be potential biomarkers in various forensic applications.

### 3.2. Small RNA Expression Profiles

The accurate quantification of small RNA is important to characterize small RNA profiles. To obtain more real and accurate small RNA profiles, the adenylated 3′ adapters with UMI were ligated to the small RNAs before the reverse transcription. This is a key step to create a unique identification for each molecule, and some effects from PCR amplification bias, for example, uneven sampling of the starting template, were identified and corrected [42]. Compared with the methods of counting reads directly, UMI could provide the absolute quantified estimates of small RNA expression [43], which was different from most previous transcriptome studies on forensic stains [37,44]. Furthermore, we utilized the BGISEQ-500 as a sequencing platform, which could minimize the replication errors and make the read distributions more even [45,46].

In this study, after normalization and filtering, 1394 miRNAs (including 236 novel miRNAs), 3157 piRNAs, 636 snoRNAs, and 17 snRNAs were detected, reaching a total of 5204 small RNAs (EXP > 0 was counted). The percentage of different RNA classes for each group is illustrated in Figure 3A. For every group, miRNA was the main class, and the second abundant class, sno/snRNA class, accounted for approximately 30%, except for the SE group (sno/snRNA class occupies 12.3%). While the piRNA was the absolute majority class in the SE group, the VB group had the least piRNA, which was consistent with Wang’s study [10]. This is because piRNA plays an essential role in germline cells [47]. Furthermore, common RNAs shared among different groups and group-specific RNAs were investigated and illustrated in Figure 3B by intersection distribution. The SE group had much more RNA (3980) compared to the other five groups (averagely 1635), and the SK group had the least RNAs (1515). Meanwhile, SE also had the most unique small RNAs (2291), while the VS group had the least (56).

In our study, the read count of small RNA was standardized to the EXP value for follow-up analysis. After standardization, the EXP value can be used to compare the different groups and samples directly. Based on this value, the heatmap (Appendix A) presented the expression level of the whole small RNAs in all of the samples. We found that the SE and VB groups showed relatively high expression (in red), perhaps due to abundant piRNAs and miRNAs, respectively, while the performance of the SK and SA groups were scattered. In this study, the small RNAs were divided into three grades (low expression: 0 < EXP ≤ 1; moderate expression: 1 < EXP < 10; high expression: EXP ≥ 10) based on their expression level. As shown in Figure 4A, the number of RNAs that were expressed at a high expression level was lowest in the SK group and highest in the VB group. One possible reason is that venous blood carries more important physiological functions while skin does not. Furthermore, we found that the expression abundance of SE004 was different from other samples in the SE group, having many small RNAs with a “moderate expression level”. Individual variability or other health factors could be the reasons for this.

To further compare the expression level distribution, each RNA with an average expression of more than 1 EXP value was presented in Figure 4B, and the differences within and between groups were observed. Similar to the results of Figure 4A and Appendix A, different groups demonstrated various expression level distributions, though some groups showed a relatively consistent expression, for example, the VB group. However, on the whole, the expression level of the skin group was much lower than the remaining groups.

In terms of expression level, the repeatability of the library construction and sequencing was studied by adding a technical duplication (VB006) of a blood sample (VB002). The Pearson correlation coefficient of these two samples was 0.9794. This indicates that their expression was extremely similar and they could be considered to be from the same sample. In other words, the quality of the library construction and sequencing were reliable. In all of the subsequent analyses of the VB group, VB006 was not included.

### 3.3. Body Fluids/Tissue-Specific RNAs

The differential RNA analysis, which intuitively reflects the biological phenomenon behind it, is a common task in medicine. For forensic applications, differentially expressed RNAs are the ideal source of a specific marker for BFID in crime investigation, and the discovery of these kinds of markers are of great significance. In this study, 15 pairwise groups (VS_MB, VB_MB, SE_MB, VS_SA, SK_SA, SK_MB, SK_SE, VB_SA, SA_MB, VB_SK, VB_SE, SE_SA, VS_VB, VS_SK, VS_SE) were set for differential RNA analysis. The results were visualized in the volcano plots (Appendix A). Generally, the RSMs defined for picking forensic body fluids/tissue out of a pair of body fluids were selected from these significantly differentially expressed RNAs. All RSMs of 15 pairwise groups are summarized in Appendix A and offers an inventory of potential RNA markers for small-scale forensic body fluids and skin tissue identification.

Furthermore, 90 ASMs that were specifically expressed in a certain type of body fluid/tissue were screened and are shown in Table 2, and some piRNAs and snoRNAs were reported first. Among the six groups, the VB group had the most ASM (a total of 41 with 19 miRNAs, eight piRNAs, and 14 snoRNAs), followed by the SE group (a total of 25 with eight miRNAs, 11 piRNAs, and six snoRNAs) and the SK group (10 snoRNAs). The MB, SA, and VS groups had less than ten ASMs. Thereinto, a novel miRNA named novel-miR254-3p (temporary name, precursor sequence predicted: CCACCACGCCUGGCCUAAGAGUGUGUUUCUUAAAGGUAAAGACUUGAGGUCGGACAUGGUGGCU; mature sequence predicted: UUGAGGUCGGACAUGGUGGCU) was regarded as the ASM of the MB group. Unfortunately, we did not find ASMs in the SA group, which may be attributed to the low RNA content to some extent. However, in the ASM filtering analysis, we noticed that the expression level and the specificity of some markers were sometimes incompatible. Ideally, it will be a useful marker to identify tissue origin if it has a higher expression level and specificity, in which the former is detection-oriented and the latter is efficiency-oriented. For better characterization and selection of RNA markers, both the expression and specificity were set hierarchically to levels I to IV to rate the described ASMs. The grade information of the top 10 potential ASMs is presented in Table 3. Taking miR-144-3p, an ASM of the MB group, as an example, its specificity reached level IV in four of the five pairwise groups (MB_SA, MB_SE, MB_SK, MB_VB, MB_VS) and reached level III in the remaining group. With an average expression level of 376.19 (EXP value) in the MB group, miR-144-3p was ranked as level III (“+”). In this example, the marker had high reliability in MB-identification due to the high specificity and expression level.

To verify the discriminating power of ASMs, we conducted hierarchical clustering analysis for all samples based on Euclidean distance. As shown in Figure 5A, although there were some overlaps between samples with higher homogeneity, relatively clear differential patterns were observed, and most samples from the identical group were gathered together. First, all samples in the VB and SE groups clustered together, demonstrating that ASMs in the two groups had high discriminating power. For the remaining four groups, different extents of misclassification were observed, perhaps because all of these body fluids/tissue involve epithelial cells. Furthermore, an unbalanced number of ASMs could also be an important influencing factor for the clustering results. In the following analysis, when the unsupervised t-SNE method was used to test the performance of the ASMs, a similar tendency was observed (Figure 5B). The samples from the VB and SE groups were clearly separated. Samples from SA could also be grouped together but they, as a whole, mixed with VS and MB. For the SK group, four samples could cluster correctly except for one sample mixed with MB.

Up until now, many body fluids/tissue-specific small RNAs have been reported for the six stains, most of which are miRNA (Table 2). Some potential markers described in this study overlapped with the publicly reported ones such as miR-144 and miR-214 for the MB group; miR-891, miR-888 for the SE group; miR-126, miR-484 for the VB group (Table 2), all of which have been verified repeatedly [17,18,52,63]. However, there were also some published miRNA markers that did not meet our criteria and were filtered out, resulting in no miRNA biomarkers for the SA and SK groups. Their expression level is shown in Appendix A. Further analyses of the expression level of these miRNAs revealed that low expression level or low fold changes of expression were the main cause. Some previously reported biomarkers could be false positive because of bias from PCR or sequencing. Another reason that could not be ignored is individual variability.

We also noticed that most of the candidate miRNA markers for BFID did not coincide among the published studies. For instance, the miR-203, which was regarded as a SA-specific marker in previous studies [17,25,52,63], was selected as the VS marker in this study (Figure 6A). Since some studies showed that miR-203 and miR-205 were epithelial cell-specific markers [64], theoretically, they could be detected in tissues involving epithelial cells such as saliva and vaginal secretion. In fact, miR-203 (miR-203a-3p and miR-203b-5p) could be detected in almost all samples, although their expression level differed among samples (Figure 6A). For instance, the average expression level of miR-203a-3p in the VS group was nearly 7 times that in the SA group. The degree of deviation was comparable to that reported in the study by Sirker et al. [25]. Therefore, miR-203 should be an ASM for VS identification rather than SA identification. Similarly, miR-205, which was reported to be a marker for SA [27,52,53] or SK identification [39], was not listed as the AMSs for either SA or SK in this study. As shown in Figure 6B, miR-205 had a high expression level in four of the six body fluids/tissue (MB, SA, SE, and VS), demonstrating that the specificity of miR-205 was insufficient for the SA group. Regarding the miR-144 marker, miR-144-3p and miR-144-5p were regarded as ASMs of the MB in this study (Table 2), which were consistent with the studies by Sauer [25] and Zubakov [48]. However, in the studies by Wang et.al [27] and He et al. [17], miR-144-3p and miR-144-5p had VB specificity. However, it was worth noting that the study from Bamberg et al. [65] revealed that miR-144-3p was not able to identify VB and MB. In our study, miR-144-5p and miR-144-3p had high expression levels in both the MB and VB groups (Figure 6C), especially miR-144-3p in the MB. Aside from the above markers, miR-484 was selected as an ASM of the VB, which was also reported by Park et al. [54]. However, in some studies, miR-484 was used as a reference gene [56,60]. In our study, miR-484 had an unbalanced expression in six stains (Figure 6D), with a high expression level in the VB group. A similar trend was reported by Bamberg et al. [65], except for the moderate expression observed in semen in this study. A conflict was observed in the miR-135b marker (Figure 6E), which was thought to be SA-specific [51] or SE-specific [48,50,55] in previous studies. In our study, however, it was not specific to any of the six stains. Overall, although miRNA was a potential marker for forensic BFID, the influence of individual variability, detection and screening methods, etc. may result in inconsistency, which needs to be explored further to prevent application obstacles.

Generally, piRNA is mainly expressed in semen. There were 11 new piRNA ASMs in the SE group (Table 2). Surprisingly, eight new piRNAs were also identified as ASMs in the VB group. piR-55521, which was reported as a potential semen-specific marker [66], was not observed in our study. Furthermore, among the body fluid identification markers for discriminating VB and MB (piR-27622, piR-001207, piR-027493), and SA and VS (piR-027493, piR-026591) reported by Wang et al. [10], only piR-001207 was detected in our study, but not selected as an ASM due to non-significant expression difference (log_2_ FC: 1.30; *q*-value: 0.52) between VB and MB. Although piRNAs were abundant in the reproductive system, studies have also shown their expression in other body fluids or tissue such as menstrual blood, saliva, skin, venous blood, and vaginal secretions [10,66]. In this study, two piRNA ASMs were found in the VS group, less than that of the reported and identified miRNA markers for BFID. Therefore, more studies related to piRNA need to be performed to verify its application value in BFID.

In this study, snoRNA ASMs were identified in four of the six body fluids/tissue. In particular, the number of snoRNA ASMs reached 14 and 10 for the VB and SK groups, respectively (Table 2). This is unexpected because, conventionally, sno/snRNAs were thought to be conservative among different tissues due to their fundamental biological function. In recent years, there have indeed been some studies showing different observations. In 2020, MaCann et al. [67] reported cell-type-specific H/ACA snoRNA abundance in the mouse, and some snoRNAs levels were regulated specifically during differentiation. In addition, Tomaž Bratkovič et al. [68] found that SNORD115 and SNORD116 were brain-specific snoRNAs. Therefore, snoRNA could have time and space specificity, and now that the sno/snRNA have been discovered as ASMs for BFID for the first time here, they are worthy of deeper verification and research.

### 3.4. Reference Genes

The selection of an optimal reference gene is a prerequisite for the accurate normalization of gene expression. Ideally, reference genes should be stable at the expression level in various tissues and under different developmental stages and external conditions. Thus, for forensic BFID, its stability of expression in different body fluids/tissue is fairly critical. In the present study, a total of 11 reference genes (Figure 7) including nine miRNAs, one snRNA, and one piRNA were found in six stains (Table 2). Among them, RF00026 (U6) [15,50], let-7a [59], let-7i [9], let-7g [9], miR-22-3p [57], miR-26a-5p [58], and miR-451a [9] have been reported, and let-7f-5p, let-7b-5p, miR-181a-5p, piR-020829 were newly identified. However, as illustrated in Figure 7, although there were relatively similar expression profiles between samples of the same fluid or tissue type, samples of different body fluids or tissues did not exhibit consistent expression levels. This phenomenon was also observed in another study when Wang et al. evaluated the stability of 18 small RNA reference genes in five common types of body fluids (i.e., VB, MB, SA, SE, and VS) [69]. It was possible that due to the heterogeneity and complexity of small RNA expression in different body fluids or tissues, it was almost impossible to find universal small RNAs, which could be applied in all types of samples as the ideal reference gene [69]. Hence, the optimal reference gene should be selected according to the target sample types. For reference genes that are screened under certain reasonable criteria, although expression variations are inevitable, just as in this study (Figure 7), they can still be used as reference genes.

## 4. Conclusions

In recent years, many studies on miRNA have been conducted for body fluids and skin tissue identification, either based on RT-PCR or MPS. In our study, six types of forensically relevant biological samples were studied by performing small RNA massively parallel sequencing using the UMI library on the BGI platform. Compared with previous studies, we comprehensively characterized the small RNA profiles of menstrual blood, saliva, semen, skin, venous blood, and vaginal secretion based on the UMI quantitative method. The results showed that the influence on the sequencing quality by RIN value was limited. Hence, samples with low RIN are also recommended to conduct small RNA sequencing. After data cleaning and standardizing, a total of 5204 small RNAs were detected including 1394 miRNAs (including 236 novel miRNAs), 3157 piRNAs, 636 snoRNAs, and 17 snRNAs. We focused on the characterization of their profiles including the small RNA class and expression level in each body fluid/tissue group. Subsequently, we defined two classes of body fluid/tissue-specific markers (RSMs and ASMs) for applications in different forensic scenarios. As a result, many RSMs and ASMs for the six types of body fluids/tissue were screened, among which some were new small RNA markers. Given the variation in the expression and specificity level of ASMs, they were set hierarchically to levels I to IV, which contributed to the selection of ideal BFID markers. Meanwhile, we also found some inconsistent markers between our results and other studies, a phenomenon that was reported in previous publications. Nonetheless, our study expanded the number of available candidate small RNA markers for conducting BFID in forensic studies and applications. In addition, a total of 11 reference genes were predicted based on MPS data, which could be used as potential markers to standardize gene expression. In the future, more samples should be investigated to explore and verify the RSMs, ASMs, and reference-gene markers.

## Figures and Tables

**Figure 1 genes-13-01530-f001:**
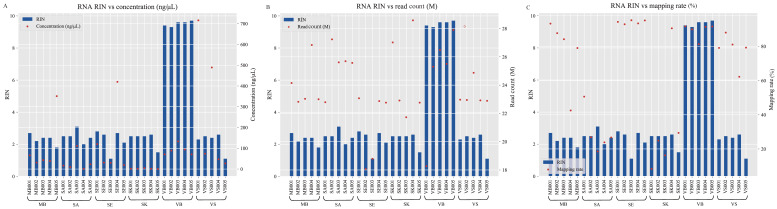
A comparison of the RIN value with the corresponding concentration (**A**), read count (**B**), and mapping rate (**C**).

**Figure 2 genes-13-01530-f002:**
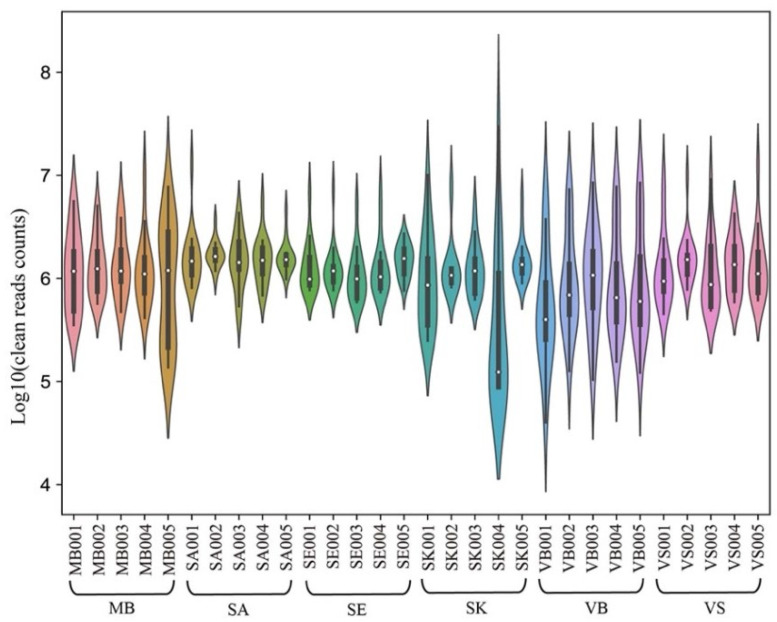
The counts distribution of the clean small RNA reads in five forensic body fluids and one skin tissue.

**Figure 3 genes-13-01530-f003:**
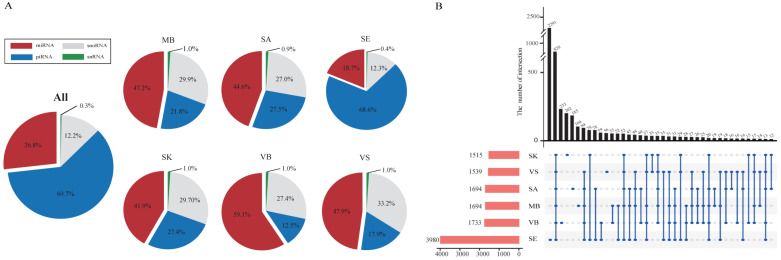
Quantity distribution of the small RNAs in six groups. (**A**) RNA class percentage in all groups; (**B**) intersection distribution of RNAs between different groups. MB = menstrual blood; SA = saliva; SE = semen; SK = skin; VB = venous blood; VS = vaginal secretion.

**Figure 4 genes-13-01530-f004:**
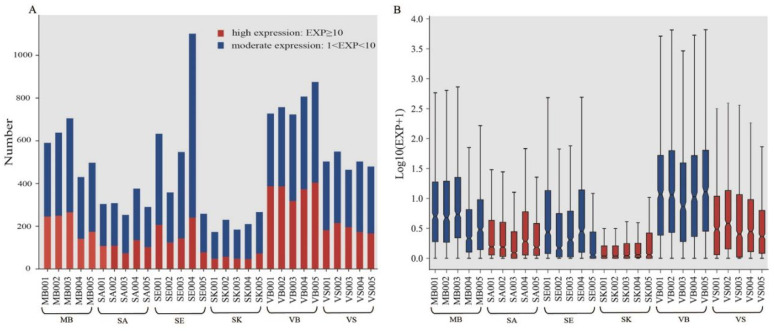
The expression distribution in 30 forensic related samples. (**A**) The number of markers with moderate and high expression level; (**B**) expression level of each sample (average EXP > 1 is presented).

**Figure 5 genes-13-01530-f005:**
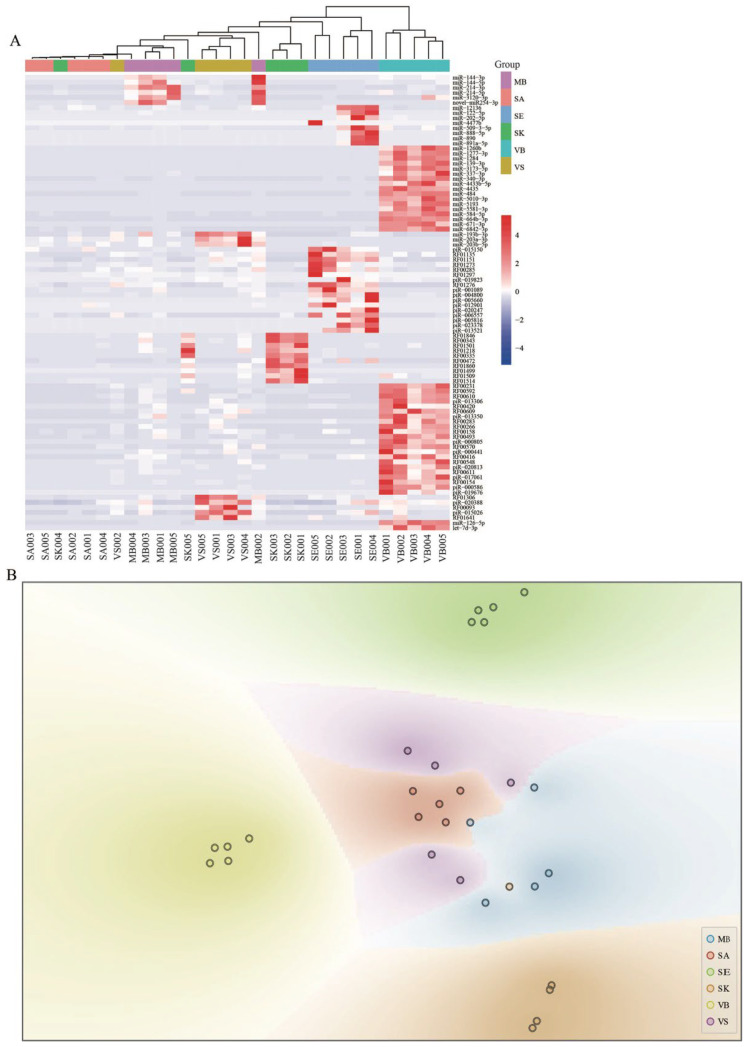
The expression patterns of the 90 ASMs screened in six forensic groups revealed by hierarchical clustering based on Euclidean distance (**A**) and unsupervised t-SNE method (**B**).

**Figure 6 genes-13-01530-f006:**
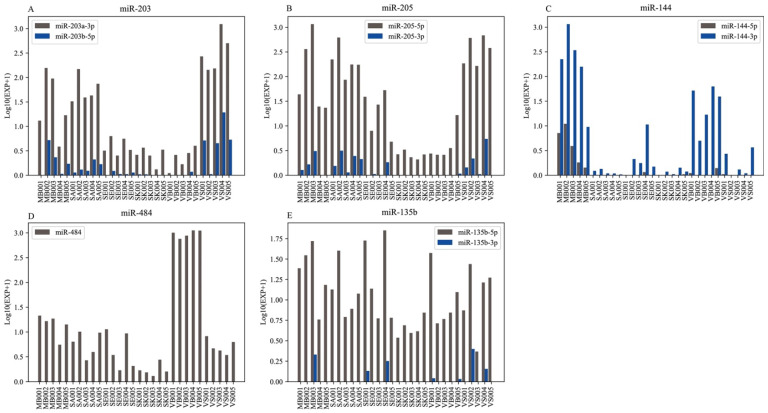
A comparison of the expression level of five miRNAs in all samples. (**A**) miR-203; (**B**) miR-205; (**C**) miR-144; (**D**) miR-484; (**E**) miR-135b.

**Figure 7 genes-13-01530-f007:**
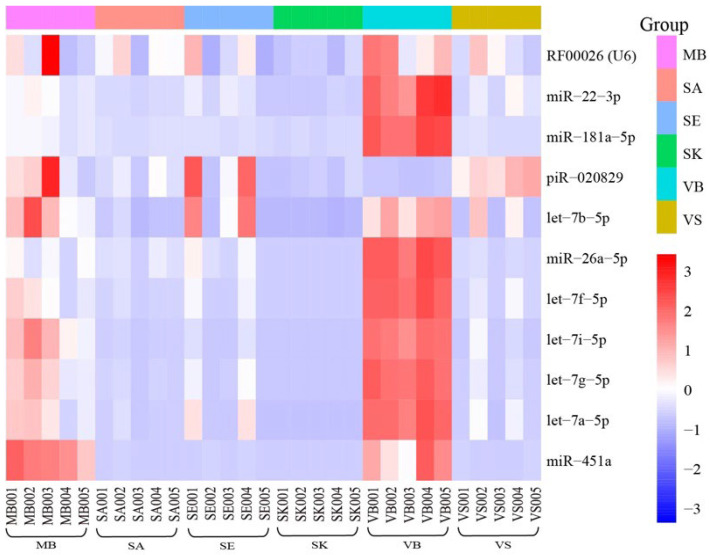
The expression level of the 11 reference genes screened in this study.

**Table 1 genes-13-01530-t001:** The quality and sequencing information of the different groups of samples.

Group	Mean Concentration (ng/μL)	Mean RIN	Mean Q20 (%)	Mean Read Count (M)	Mean Mapping (%)
MB	106.00	2.30	98.78	23.97	77.34
SA	32.23	2.50	98.78	25.38	29.31
SE	124.00	2.30	98.80	21.11	94.21
SK	1.90	2.32	98.76	24.60	33.90
VB	91.60	9.52	98.82	24.69	89.18
VS	270.60	2.18	98.92	23.32	77.94

**Table 2 genes-13-01530-t002:** The ASMs and reference genes screened in the five body fluids and one skin tissue.

Group	From	Markers
MB	Published data	**miR-144** [25,48], **miR-214** [26,49], miR-185 [25,48], miR-412 [50], miR-451 [50], miR-1246 [9]
	This study	**miR-144-3p**, **miR-144-5p**, **miR-214-3p**, **miR-214-5p**, miR-3120-3p, novel-miR254-3p
SA	Published data	miR-26a [51], miR-96 [51], miR-135b [51], miR-182 [51], miR-200c [27,52], miR-203 [9,27,52], miR-205 [27,50,52,53], miR-208b [48], miR-381 [51], miR-431 [51], miR-450b-5p [51], miR-518c [48], miR-583 [48], miR-622 [51], miR-658 [50], miR-1228 [51], miR-223 [27,54], miR-145 [51,54], miR-141 [27,51], miR-375 [27], miR-34a [27], let-7c [27], miR-22 [27], miR-27a,b [27],miR-23a,b [27], miR-125b [27],miR-99a [27], miR-29a,b [27], miR-210 [27]
	This study	-
SE	Published data	**miR-891a** [9,25,48,49], **miR-888** [49], miR-10a [48], miR-10b [48,50,55], miR-17 [51], miR-29b-2 [51], miR-135b [48,50,55], miR-340 [51], miR-380 [51], miR-507 [48], miR-508-5p [51], miR-644 [51], miR-943 [48], miR-2392 [48,54], miR-3197 [54], miR-26b [9]
	This study	**miR-891a-5p**, **miR-888-5p**, miR-12136, miR-122-5p, miR-202-5p, miR-4477b, miR-509-3-5p, miR-890, piR-001089, piR-004800, piR-005660, piR-005816, piR-006557, piR-012901, piR-013521, piR-015150, piR-019823, piR-020247, piR-023378, RF00285, RF01135, RF01151, RF01273, RF01276, RF01297
SK	Published data	miR-203a-3p [39], miR-205-5p [39], miR-139 [56], miR-494 [56], miR-3169 [56]
	This study	RF00335, RF00343, RF00472, RF01218, RF01499, RF01501, RF01509, RF01514, RF01846, RF01860
VB	Published data	**miR-484** [54], **miR-126** [52], miR-16 [48,49,50,51], miR-20a [48], miR-106a [48], miR-150 [52], miR-185 [48], miR-451 [27,48,50,52,53], miR-182 [54], miR-144-3p [17,27], miR-200b [9],miR-486 [27,49], miR-16 [27], miR-126 [27]
	This study	**miR-484**, **miR-126-5p**, let-7d-3p, miR-1260b, miR-1277-3p, miR-1284, miR-139-3p, miR-3173-5p, miR-337-3p, miR-340-3p, miR-4433b-5p, miR-4435, miR-5010-3p, miR-5193, miR-5581-3p, miR-584-5p, miR-664b-3p, miR-671-3p, miR-6842-3p, piR-000441, piR-000586, piR-000805, piR-013306, piR-013350, piR-017061, piR-019676, piR-020813, RF00154, RF00158, RF00231, RF00266, RF00283, RF00416, RF00420, RF00493, RF00548, RF00570, RF00592, RF00609, RF00610, RF00611
VS	Published data	miR-124a [50,52], miR-372 [50,52], miR-617 [48], miR-654-5q [54],miR-155-5p [25], miR-1260b [27,49], miR-654-5p [27]
	This study	miR-193b-3p, miR-203a-3p, miR-203b-5p, piR-015026, piR-020388, RF00093, RF01306, RF01641
Reference gene	Published data	**U6** [15], **miR-451a** [9], **miR-22-3p** [57], **miR-26a-5p** [58], **let-7a** [59], **let-7g** [9], **let-7i** [9], miR-484 [56,60], 5S-rRNA [60], miR-92a-3p [60], miR92 [61], miR374 [61], miR-26b [56], miR-92 [61], miR-93 [59,62], miR-191 [59,62], miR-374 [61], miR-423 [61], RNU6b [50], RNU19 [62], RNU24 [59], RNU38B [62], RNU43 [62], RNU48 [59], RNU49 [62], RNU66 [62]
	This study	**RF00026(U6)**, **miR-451a**, **miR-22-3p**, **miR-26a-5p**, **let-7a-5p let-7g-5p**, **let-7i-5p**, let-7f-5p, let-7b-5p, miR-181a-5p, piR-020829

Note: Bold case indicates common markers between the published studies and this study; “-” means no markers are available.

**Table 3 genes-13-01530-t003:** The specificity and expression level of the top 10 ASMs screened in five body fluids/tissue.

Group	Small RNA	Specificity				Expression Level			
		I	II	III	IV	I	II	III	IV
MB	miR-144-3p			1	4			+	
	miR-214-3p				5		+		
	miR-214-5p				5		+		
	miR-3120-3p				5	++			
	miR-144-5p				5	++			
	novel-miR254-3p				5	+			
SE	miR-12136			2	3			+	
	piR_005660				5			+	
	miR-890				5			+	
	piR-015150		1	1	3			+	
	miR-888-5p				5			+	
	piR-019823		1		4		+++		
	piR_020247				5		++		
	RF01297				5		++		
	RF01151	1		2	2		+		
	miR-122-5p				5		+		
SK	RF00335				5			+	
	RF01846				5			+	
	RF01860				5		++		
	RF01499				5		++		
	RF01509			1	4		+		
	RF01501				5	+++			
	RF01218			1	4	++			
	RF01514				5	++			
	RF00343				5	++			
	RF00472		1		4	++			
VB	miR-584-5p				5				+
	miR-484				5			+++	
	let-7d-3p				5			+++	
	miR-337-3p				5			++	
	miR-126-5p				5			++	
	RF00231			2	3			+	
	RF00610			2	3			+	
	miR-1284				5			+	
	piR-013306			2	3			+	
	piR-000805			1	4		+++		
VS	miR-193b-3p		1	1	3			++	
	miR-203a-3p		1	1	3			++	
	RF01306			2	3			+	
	RF00093			2	3			+	
	piR-020388		4	1				+	
	piR-015026			3	2		+		
	RF01641			2	3	+++			
	miR-203b-5p		1	1	3	++			

Note: Specificity level was divided to level I (1 < log_2_FC ≤ 2), level II (2 < log_2_FC ≤ 3), level III (3 < log_2_FC ≤ 4), and level IV (log_2_FC > 4). The content represents the number of comparison groups at the corresponding specificity level. Expression level was divided into level I (1 < EXP ≤ 10), level II (10 < EXP ≤ 100), level III (100 < EXP ≤ 1000), level IV (EXP >1000). For expression level groups I–III, they were further divided into different subgroups, with “+” is the expression level that was more than the lower bound of the interval and less than the 1/3 interval; “++” is the expression level that was more than the 1/3 interval and less than the 2/3 interval; and “+++” is the expression level that was more than the 2/3 interval and less than the upper bound of the interval. For expression level IV, no further division was applied. A detailed example is presented in the text. Since no ASM was screened in the SA group, a total of five body fluids/tissue were illustrated here.

## Data Availability

The data presented in this study are available on request from the corresponding author. The data are not publicly available due to the restriction of privacy and law.

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
