# Peer review of "A Comprehensive Characterization of Small RNA Profiles by Massively Parallel Sequencing in Six Forensic Body Fluids/Tissue"

_genes, 2022, doi:10.3390/genes13091530_

Round 1

Reviewer 1 Report

The topic investigated is quite interesting for the forensic genetics community. As stated by the Authors, understanding of small RNA profiles in forensic stains can be applied in different forensic scenarios. The limit of the manuscript could be that the number of tested samples is low and it should be increased, for example for an other manuscript.

Reviewer 2 Report

Review comments:

In this manuscript, the authors characterised different types of small noncoding RNA classes of 5 different body fluids and skin tissue by MPS, and identified different groups of RNAs that may be classed as “relative” or “absolute” specific markers to be utilised in forensic body fluid identification (BFID). Notably, this manuscript investigated piRNAs, snoRNAs and snRNAs which have been poorly studied for BFID applications and identified candidates in these RNA classes that may offer greater reliability in BFID in comparison to miRNAs.

I would like to commend the authors on the amount of work completed, and I have enjoyed reviewing this manuscript. Given the complexity of the data presented, I appreciated the examples in text to explain the concepts, and the effort made by the authors to convey the results. I have many questions out of my own curiosity and I also feel that based on the results presented, the authors are selling themselves a little short on the conclusion, and perhaps could be expanded further.

In the introduction page2, the snRNAs are much longer in length in comparison to the other sncRNA classes. Given the unknown nature and poor quality of samples in real casework samples, are snRNAs less suitable for BFID compared to piRNA and snoRNAs? The two classes of snRNAs, the sm-class and lsm-class, are they referring to the length?

Have the authors previously looked at the changes to RNA expression profiles over the menstrual cycle (or have there been studies previously which indicate cycle-dependent variations)?

For the small RNA library preparation and sequencing, between 200ng-1ug of input RNAs were used. I take it that for some samples where the total RNA yield was low, the target optimal RNA input could not be reached. Is that true? What is the optimal targeted input used?

Were the UMIs synthesised in the lab or which company was it from? PCR Primer cocktail / PCR mix where is that from? Some companies are indicated in the methods section but there are many missing.

In section 3.3. Table 3 is mentioned before Table 2, can this be renamed or reorganise the tables such that they are in the order as they are discussed in text.

Based on the results for the RSM/ASMs identified for the different body fluids, can the authors propose a model/guidelines for BFID on mixed stains? Can the authors recommend minimum requirements for BFID (with any statistical confidence) as to how many miRNAs/snoRNAs/piRNAs etc for detection?

The authors have shown that the miRNA results are variable and not the best candidates for BFID, which is also in agreement with previous studies investigating miRNA expression for BFID, are the authors able to provide recommendations for more reliable and consistent BFID (i.e. more work on piRNA, snoRNA and snRNAs)?

The reference genes section seems to be a small add on – can the authors expand on it? I understand the scope of this manuscript is reporting on the RNA expression profile of the 30 samples, but it will be interesting to determine the utility of these reference genes in BFID. Have any of these candidate reference genes been applied to the sequencing results to see if they strengthen the identification of the body fluids or in mixed stains?

In Figure 7, based on the heat map for the candidate reference genes in the different body fluids tested, some of them show relatively similar expression levels between samples of the same fluid type, but they do not appear to exhibit stable expression levels between the different body fluids. Can the authors comment on the expression of these candidate reference genes based on the results in this study and whether they are suitable? The expression look quite variable from the heat maps.    

 Grammatical / phrasing comments:

Suggested phrasing/spelling underlined-

Page 2 line 47: “in forensics, miRNAs have been proposed as novel and potential forensic BFID markers………”  

Page 2 line 51: “…or chemical factors”

Page 2 line 57: “….and attracting increasing interests from forensic geneticists…”

Page 2 line 68: “…plenty of studies have been conducted to investigate sncRNAs for BFID….”

Page 2 line 74: “……forensic biological materials will be highly valuable….”

Page 6 lines 196, 197 and 202: avoid abbrieviations in text - change “didn’t” to did not.

Page 6 line 208: when referring to “unequal identification for each molecule”, did the authors mean unique identification? I think unique is a better word.

Page 6 line 212: replace “researches” to “studies”

Page 11 line 331: replace “checking” to “analyses”

Page 11 line 334: awkward phrasing - replace “individual differences” to “individual variability with RNA expression profile”.

Page 13 line 364: replace “controversial” with “conflict”

Page 13 line 386: replace “out of expectation” with “unexpected”
